# Prediction of Factors Affecting Mobility in Patients with Stroke and Finding the Mediation Effect of Balance on Mobility: A Cross-Sectional Study

**DOI:** 10.3390/ijerph192416612

**Published:** 2022-12-10

**Authors:** Fayaz Khan, Sami Abusharha, Aljowhara Alfuraidy, Khadeeja Nimatallah, Raghad Almalki, Rafa’a Basaffar, Mawada Mirdad, Mohamed Faisal Chevidikunnan, Reem Basuodan

**Affiliations:** 1Department of Physical Therapy, Faculty of Medical Rehabilitation Sciences, King Abdulaziz University, Jeddah 21589, Saudi Arabia; 2Department of Occupational Therapy, Faculty of Medical Rehabilitation Sciences, King Abdulaziz University, Jeddah 21589, Saudi Arabia; 3Department of Rehabilitation Sciences, College of Health and Rehabilitation Sciences, Princess Nourah bint Abdulrahman University, P.O. Box 84428, Riyadh 11671, Saudi Arabia

**Keywords:** stroke, gait, balance, mobility, lower limb, prediction, mediation

## Abstract

(1) Background: Regaining mobility after stroke is essential to facilitate patient independency in activities of daily living. Predicting post-stroke mobility is clinically important and plays a significant part in rehabilitation programs. The purpose of this study is to find the factors affecting mobility in patients with stroke and to analyze the mediation effect of balance on mobility. (2) Methods: This cross-sectional study included forty-one patients with stroke averaging an age of 57.2 ± 88.6. The Rivermead Mobility Index (RMI) was used for measuring the mobility, Timed Up and Go (TUG) to measure the walking speed, Berg Balance Scale (BBS) to assess the balance and a handheld dynamometer (HHD) was used for measuring the isometric strength of the ankle and knee. (3) Results: In regression analysis balance (β=0.58; p≤ 0.0001) and walking speed (β=−0.27; p=0.04) were the significant factors predicting mobility. (4) Conclusions: Balance and gait speed were the factors that influenced mobility in stroke patients, indicating the utility of measuring these aspects in order to provide appropriate rehabilitation programs.

## 1. Introduction

Stroke is the world’s second leading cause of death and the most common reason for disability among adults [1]. A stroke is described as ‘a neurological deficit attributed to an acute focal injury of the central nervous system (CNS) by a vascular cause’ [2]. Fifty percent of stroke survivors experience long-term disability with more than a third of them requiring assistance in the performance of daily life activities [3]. The most common disability after a stroke is motor impairment of the contralateral limbs, which affects more than 80% of stroke patients in the acute phase and 40% chronically [4]. Post-stroke motor impairment can lead to reduced mobility, which is a significant hardship for stroke victims, their families, and health-care providers [5].

During rehabilitation, people with stroke prioritize regaining their mobility due to its importance in being independent during day-to-day activities [6]. Predicting mobility outcomes is necessary for health care providers to accurately establish achievable rehabilitation milestones, enable effective discharge planning, and to identify the need for community help and home modification [7,8]. According to the International Classification of Functioning, Disability, and Health (ICF), mobility is defined as an individual’s capacity to move around effectively in his environment [9]. In the context of stroke rehabilitation, mobilization has been described as ‘out of bed physical activity’, this may include tasks such as transferring on or off the toilet, sitting up, getting out of bed, standing, and walking [10].

Several studies have been conducted in order to identify predictors of mobility and functional outcomes among stroke survivors in the chronic and acute phases [6,11,12,13]. The development of prognostic models and the identification of factors that predict post-stroke functional outcomes have been the focus of numerous studies. However, motor recovery after a stroke is a complex, dynamic, and multifaceted process that results from the interplay of various genetic, physiologic, and sociodemographic factors. Some factors, such as comorbidities, access to acute care, therapy, and age, may have an impact on recovery. Nonetheless, the severity of the stroke remains the primary factor that predicts recovery from stroke. This assumption is backed by the fact that the majority of the variables shown to be predictive of recovery are side effects of stroke severity. These side effects include unconsciousness, disorientation, paralysis, urinary incontinence, balance, gait disturbances, and muscle weakness. Many studies have been conducted to determine which factor best predicts future functional mobility recovery. One factor that researchers have thoroughly explored is balance. Balance performance is commonly identified as a potential predictor of long-term mobility after a stroke; patients with impaired balance recover less functional mobility than persons with normal balance performance [11,13,14]. Regaining balance is viewed as a crucial component of stroke rehabilitation and predicting mobility outcomes using balance can provide healthcare providers with significant clinical insight. Despite the prognostic value of balance, no study has attempted to explore the mediation effect of balance on mobility.

Recovery of walking ability is essential to the rehabilitation of stroke victims [15]. One in two stroke victims will initially be unable to walk, although they will eventually recover throughout the acute phase [16]. During the chronic stage, 80% will regain independent walking, although more than half will do so with some sort of gait limitation [16]. Gait speed is routinely used by researchers and clinicians to determine walking capacity following a stroke [17]. Speed of gait is a reliable measurement of recovery of walking ability and is frequently used on stroke patients to estimate their functional ability to ambulate both inside and outside of their home [18]. It has also been used to predict mobility in stroke survivors [15,17].

Gait and balance are integral parts of functional mobility, and their performance is aligned primarily with the lower extremities (LE) condition [19]. Stroke survivors view lower limb disability as a more significant sign of limited mobility than upper limb impairment [20]. Certain foot and ankle disorders have been found to be important predictors of balance and functional capacity among older adults [21]. Stroke is commonly associated with dramatic deterioration in lower limb muscle mass, causing decreased balance and an inability to maintain mobility, especially in gait performance [22].

This study intends to investigate the correlation and predictability of factors associated with mobility among a population of sub-acute and chronic stroke survivors and to find the mediation effect of balance on mobility which, to our knowledge, has not been investigated before.

## 2. Materials and Methods

### 2.1. Study Design

This cross-sectional study was designed to find the various factors affecting mobility in patients with stroke and the influence of balance as a mediator on mobility. Assessment sessions were conducted before participants’ routine physical therapy appointments. Participants were assessed by trained physical therapists for the following: balance by Berg Balance Scale (BBS), functional mobility by Rivermead Mobility Index (RMI), gait speed by Timed Up and Go (TUG), and lower extremity strength by a handheld dynamometer. Informed consent was obtained from the participants before they were enrolled in the study and the ethical approval was obtained from the Centre of Excellence in Genomic Medical Research (04-CEGMR-Bioeth-2019), approved by the National Committee of Bioethics (KACST: HA-02-J-003).

### 2.2. Participants

Forty-one patients with stroke were recruited for the study from department of physical therapy at King Abdulaziz University Hospital, Department of physical therapy, Abdullatif Jameel Hospital, and from the Department of physical therapy in King Fahad Hospital. Participants were enrolled in the study in the period of June 2019–December 2019. Inclusion criteria was based on: (1) patients with a duration of ≥2 months since stroke onset; (2) patients can walk independently for 10 m with or without assistive device; (3) patients should be able to follow instructions; (4) adult patients who are above 20 years of age; (5) received a diagnosis of ischemic brain injury or intracerebral hemorrhage by MRI or CT. Exclusion criteria included the following: (1) patients with orthopedic, cardiovascular, and pulmonary conditions that may restrict the assessment; (2) patients with bilateral stroke; (3) patient who were involved in drug studies or other clinical trials. Basic demographic characteristics of the study subjects are outlined in Table 1.

### 2.3. Outcome Measures

#### 2.3.1. Rivermead Mobility Index (RMI)

The Rivermead Mobility Index (RMI) is a 15-item standardized assessment that measures functional mobility in gait, balance, and transfers [23]. Performance is rated by a dichotomous (yes/no) scale, yes equals ‘1’ and no equals ‘0’. The score maximum is 15, with higher scores indicating better functional mobility. 14 items are related to self-reported performance and one item is assessed by the rater through direct observation [23].

#### 2.3.2. Berg Balance Scale (BBS)

The BBS is a 14-item scale that measures the subject’s ability to maintain balance during a sequence of different tasks [24]. Items are rated by an ordinal scale which ranges from 0–4, with 0 indicating lowest level of function (dependency or inability to perform task) and 4 indicating highest level of function (full independency during task performance). Scores of each task are added together to obtain the total score of the BBS with the highest possible score being 56.

#### 2.3.3. Time Up and GO (TUG)

TUG is a performance-based measure that assess functional mobility [25]. In the test, the subjects stand up from a chair, walk for 3.0 m at a normal pace, turn around after reaching the 3.0-m mark, walk back to the initial point, sit down on the chair. The time it takes the subject to complete the test determines the test’s score. A cutoff value of 13.5 s or longer indicate higher fall risk among older adults. 

#### 2.3.4. Handheld Dynamometer (HHD)

Handheld dynamometer (Dynatronics Corporation) was used to measure the ankle and knee strength of the stroke affected lower extremity, it was wrapped with a pad to prevent the pain the patient may feel by pressing the lower leg and metatarsals against the device. It measures the force in kilograms. The assessment for ankle strength was taken in supine position by holding the dynamometer on the patient’s plantar aspect at the metatarsal level and asking them to push downward to test for the ankle plantar flexors, and for the dorsiflexors by holding the dynamometer in the dorsal aspect of the metatarsal and asking the patient again to push upward. Knee strength was assessed by asking the patient to assume a sitting position while holding the dynamometer in the anterior lower third of the leg and pushing forward. Knee extensors and flexors were assessed by holding the dynamometer in the posterior lower third of the leg while asking the patient to push backward.

### 2.4. Statistical Analysis

The data were analyzed using statistical software SPSS version 23 (SPSS, Inc., Chicago, IL, USA) and Graph Pad Prism version 7.0 (GraphPad Software Inc., La Jolla, CA, USA). Mean, standard deviation, percentages, median, and range were used to describe the different characteristics of the participants. Linear regression was performed to find the factors affecting community mobility keeping the RMI as the dependent variable. Mediation analysis was performed using Hayes PROCESS to determine if balance mediated the relation between walking speed and mobility. The significance was determined to be *p* ≤ 0.05.

## 3. Results

### 3.1. Descriptive Results

Seventy-six subjects with stroke were screened for the inclusion and exclusion criteria and forty-one subjects with a mean age of (57.2 ± 88.6) were included in the study. Table 1 describes the demographic and baseline characteristics of the study participants.

### 3.2. Correlations

Pearson’s correlation was performed to estimate the relation between RMI with BBS, TUG, Isokinetic ankle plantarflexion and dorsiflexion, and Isokinetic knee flexion and extension. There was a significant relation between RMI with TUG (r = −0.81; *p* ≤ 0.0001; 95% CI = −0.89 to −0.65) and between RMI with BBS at (r = 0.81; *p* < 0.0001, 95% CI = 0.67 to 0.89) (Figure 1).

### 3.3. Regression

Multiple linear regression analyses were done keeping RMI as the dependent variable to predict the influence of different independent variables (TUG, BBS, isometric ankle dorsiflexion and plantarflexion, and isometric knee extension and flexion) on RMI.

#### 3.3.1. Criteria

Fulfilling the criteria of performing multiple linear regression such as the collinearity analysis, mahals distance, and Cook’s distance, independent variables (such as BBS, TUG, isometric plantar flexion, isometric dorsiflexion, isometric knee flexion, and Isometric knee extension) were put forward for further analysis.

#### 3.3.2. Regression Analysis

Multiple linear regression showed a significant result in ANOVA with *p* ≤ 0.0001 and R^2^ = 0.8 which demonstrates that there is 80% predictability in this regression analysis inputting the result in the formula: RMI = $\beta_{0}+{\;\beta }_{1}x_{1}+{\;\beta }_{2}x_{2}+{\;\beta }_{3}x_{3}+{\;\beta }_{4}x_{4}+{\;\beta }_{5}x_{5}+{\;\beta }_{6}x_{6}$.

RMI = $5.188+0.138{\;x}_{1}+0.032{\;x}_{2}+0.113{\;x}_{3}+0.065{\;x}_{4}+0.299{\;x}_{5}+0.037x_{6}$ (Table 2).

### 3.4. Mediation by Balance

Figure 2 illustrates the findings of the mediation analysis for the mediating effect of balance in the relation between walking speed and functional mobility. In a model that included balance as a mediator, balance (BBS) was found to be significantly mediating the relationship between walking speed (TUG) and functional mobility (RMI). The total effect of TUG on RMI was (β = −0.075, *p* ≤ 0.0001, 95% CI; −0.098 to −0.053), direct effect was (β = −0.019, *p* = 0.14, 95% CI; −0.046 to 0.006), and indirect effect was (β = −0.056, *p* = 0.001, 95% CI; −0.103 to −0.034) (Figure 2).

## 4. Discussion

This study focused on developing a model to predict mobility in post-stroke patients based on balance, speed, and lower limb strength. Predicting mobility has significant consequences for the amount of care required following a stroke, and restoring mobility is the aspect that patients value the most [26]. Therefore, it is important for clinicians to assess balance in order to set realistic mobility expectations. Our findings revealed that balance (β = 0.53) had significant influence as a predictor of mobility. These results are similar to other studies that used balance as a mobility predictor. A prospective cohort study done by Ingrid et al. found that sitting balance can reliably predict mobility outcomes (β = 0.29) [27]. Another study investigating the predictive validity of the Brunel Balance Assessment (a balance outcome) compared to the Rivermead Mobility Index found that balance disability was the strongest predictor of mobility when compared to other factors such as weakness, sensation, neglect, and age [11]. However, balance was a significant factor when predicting mobility after a follow up period of 3 months, while in initial testing (acute phase) muscle weakness was the better predicting factor [11]. Our studies’ results are consistent with the general consensus on balance–mobility relation, and support the clinical belief that retraining balance ability is a key aspect of stroke rehabilitation [11,28,29,30,31]. However, in our study, we did not take falls into consideration, and some studies suggest that balance performance cannot be reliably used to predict falling, so caution should be warranted when setting assumptions of fall risk using balance as a predictor [32,33].

There is a lack of literature on the predictive ability of gait speed on functional mobility. Gait speed was found to be significantly correlated with functional mobility at (r = −0.81) and had a predictive value of (β= −0.32). This finding is consistent with other studies that measured the correlation of walking speed (TUG) and functional mobility among stroke survivors [29,34]. Gait speed was also documented to be able to predict changes in mobility over time [35]. Another study done among geriatric patients identified that community-dwelling older adults who score >12 s in TUG are more susceptible to mobility impairment and should receive early evaluation and intervention [36]. These results imply the possibility of improving functional mobility by incorporating walking speed training into the rehabilitation program for patients with stroke.

Lower extremity muscular strength is generally reduced by 34 to 62% in stroke patients compared to healthy people [19]. Gait performance was directly related to insufficient weight-bearing capacity of the most afflicted lower leg muscles [37]. In our study, we examined the correlation between the strength of ankle dorsiflexion and plantarflexion and knee extension and flexion on functional mobility. Initially, we found an acceptable correlation between lower extremity strength and mobility; however, after regression analysis, ankle dorsiflexion, knee extension, and knee flexion were found to exhibit low influence on mobility (RMI) while ankle plantarflexion remained moderately relevant in predicting perceived functional mobility with Beta value 0.33. Plantar flexion strength has been consistently determined as a significant and independent predictor of balance and mobility [21,38]. These findings may be important for vulnerable stroke patients, as plantar flexion strength has also been described as an independent predictor of falling [39,40].

Our results agree with the general consensus that knee extension is positively associated with functional capacity after a stroke [41,42]. Knee extension strength has been documented to be a strong predictor of gait speed decline and incident mobility disability [43]. However, in our study, the correlation between lower limb strength and mobility is not significant enough to be considered as a reliable predictor of functional mobility.

The Berg Balance Scale have been extensively researched in how it assists in the prediction and assessment of speed and performance of walking [24,44]. Balance insufficiency is associated with a reduction in walking capacity, especially in the speed of walking [45,46]. To our knowledge, no other studies have investigated the mediation effect of balance on mobility. In our mediation analysis, we found that the total score of the Berg Balance Scale, which represented balance performance, was confirmed as a significant mediator in the relationship between walking speed (TUG) and mobility (RMI). This association may form a potential mechanism in which walking speed can affect functional mobility among stroke patients. Speed of walking is heavily reliant on balance control due to walking being a difficult task that comprises multiple steps in which the center of mass is moved outside of the limits of stability [46]. Balance training can greatly improve gait speed and performance [47]. Based on our findings, particularly the mediation effect of balance on walking speed and functional mobility, we hypothesize that balance training may positively effect walking speed performance which, in turn, can improve functional mobility outcomes.

### Limitations

The relatively small sample size is insufficient to adequately characterize the broad stroke population, limiting the generalizability of the current findings. There was no cut-off value for the dependent variable RMI which restricted the authors’ ability to perform binary logistic regression, which could have provided us with the odds of different variables in the prediction of mobility. Future studies should incorporate outcome measures which also assess community mobility so that all the domains of mobility could be predicted. Our assumptions were made based on cross-sectional data, and we did not consider confounders such as duration of stroke onset, age, gender, side, and site of the lesion. We recommend researchers investigate these factors and build on this knowledge.

## 5. Conclusions

We found that balance and walking speed were strongly associated to functional mobility, while strength of lower limb muscles showed moderate correlation to mobility.

Independent variables of balance, gait speed, and isometric plantarflexion demonstrated good prediction of the dependent variable of functional mobility (RMI), which indicates the usefulness of assessing these factors in order to provide necessary rehabilitation programs.

Our findings revealed a mediating effect of balance on the relationship between gait speed and functional mobility among a sample of stroke survivors. This finding raises the question of whether a balance training program may improve gait speed which could lead to better mobility outcomes among stroke survivors.

## Figures and Tables

**Figure 1 ijerph-19-16612-f001:**
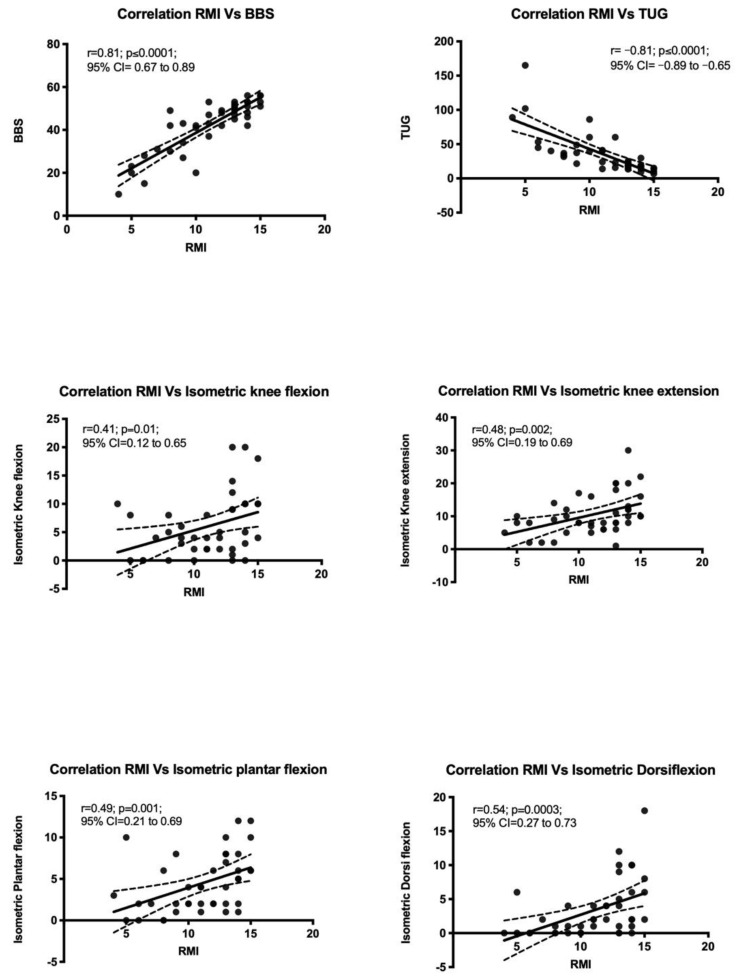
Correlation of RMI with different independent variables. TUG: Timed Up and Go test; BBS: Berg Balance Scale; RMI: Rivermead Mobility Index.

**Figure 2 ijerph-19-16612-f002:**
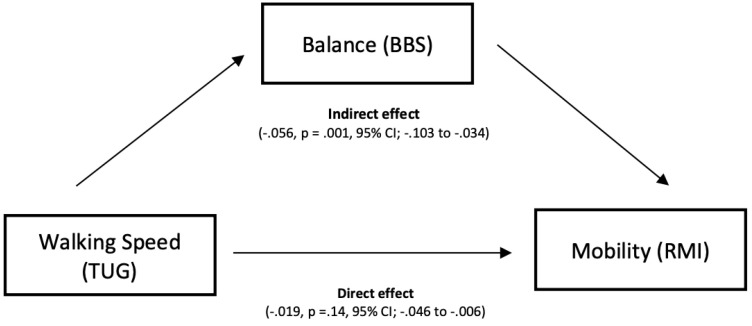
The mediating role of balance in the association between walking speed and mobility.

**Table 1 ijerph-19-16612-t001:** Basic demographic characteristics of the study subjects.

Variables	Frequency (%)
Gender (male/female)	33 (80.48)/8 (19.51)
Side of hemiplegia (left/right)	28 (68.29)/13 (31.70)
Assistive device (with/without)	18 (43.90)/23(56.09)
Diabetes mellitus (with/without)	13 (31.70)/28 (68.29)
Hypertension (with/without)	21 (51.21)/20 (48.78)
	Mean ± SD (range)
Age (y)	57.2 ± 88.6 (41 to 94)
Time since stroke (m)	18.0 ± 30.0 (2 to 168)
Ankle dorsiflexion strength (kg)	3.4 ± 4.1 (0 to 18)
Ankle plantarflexion strength (kg)	4.5 ± 3.5 (0 to 12)
Knee extension strength (kg)	10.5 ± 6.1 (1 to 30)
Knee flexion strength (kg)	6.0 ± 5.4 (0 to 20)
TUG (s)	35.0 ± 30.3 (6.75 to 165)
	Median, IQR (range)
BBS	47.14 (10 to 56)
RMI	12.5 (4 to 15)

TUG: Timed Up and Go test; BBS: Berg Balance Scale; RMI: Rivermead Mobility Index.

**Table 2 ijerph-19-16612-t002:** Unstandardized coefficients and standardized coefficients of regression analysis.

	Unstandardized Coefficients		Standardized Coefficients			95% CI	
Model	B	Std. Error	Beta	t	Sig	Lower Bound	Upper Bound
(Constant)	5.188	1.827		2.84	0.008	1.475	8.9
BBS	0.138	0.037	0.534	3.745	0.001	0.063	0.213
TUG	−0.032	0.013	−0.316	−2.409	0.022	−0.06	−0.005
Isometric Knee Flexion	−0.113	0.091	−0.195	−1.245	0.222	−0.297	0.071
Isometric Knee Extension	0.065	0.059	0.127	1.095	0.281	−0.055	0.185
Isometric Plantar Flexion	0.299	0.143	0.331	2.1	0.043	0.01	0.589
Isometric Dorsiflexion	−0.037	0.1	−0.048	−0.366	0.716	−0.24	0.167

TUG: Timed Up and Go test; BBS: Berg Balance Scale.

## Data Availability

The data presented in this study are available upon request from the corresponding author.

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
