# Peer review of "Prediction of Factors Affecting Mobility in Patients with Stroke and Finding the Mediation Effect of Balance on Mobility: A Cross-Sectional Study"

_ijerph, 2022, doi:10.3390/ijerph192416612_

Round 1

Reviewer 1 Report

Thank you for the opportunity to review this article. I attached my recommendations. 

Author Response

Response to reviewer’s comments

Reviewer 1:

Comment 1:

We recommend expanding the Introduction with the highlighting of the factors affecting mobility identified in the specialized literature with relevance for the topic of the study.

Response to comment 1:

The authors thank the reviewer for the comment and we have expanded the introduction based on the reviewer’s recommendation.

Comment 2:

We recommend identifying the novative aspects of the present study compared to previous studies.

Response to comment 2:

We added the objective of finding the mediation effect of balance to the introduction, which hasn’t been investigated before and is a novel idea in such studies.

Comment 3:

The period of the study and the important aspects of the study should be mentioned.

 Response to comment 3:

We included the study’s duration and the assessment procedure.

Comment 4:

It should be mentioned that the significant data of the study sample are analyzed in table 1.

 Response to comment 4:

We added a mention to the table.

Comment 5:

To respect the rules of the journal regarding tables and acronyms.

Response to comment 5:

We adjusted tables and acronyms according to the journal’s standards.

Comment 6:

We recommend rewriting the Discussion section to highlight the relevance of the main results identified in the current study compared to the results of previous studies, on the specific topic. In the form in which they are present in this section, the information is not systematized.

Response to comment 6:

The authors appreciate the reviewers comments and considering that the authors have re-written significant parts of the discussion to highlight the findings of our study and to compare to previous studies wherever applicable. We also adjusted wording and flow of information to provide a systemized approach.

Comment 7:

We recommend expanding the Conclusions section with 2-3 relevant ideas in relation to the identified results.

 Response to comment 7:

We added the correlation between the variables and mobility to the conclusion. The conclusion now has 3 relevant ideas.

Reviewer 2 Report

The authors present the manuscript: "Predicting factors affecting mobility in stroke patients and finding the mediating effect of balance on mobility; a cross-sectional study". I have a few thoughts on this.

Introduction: well explained and concise.

Line 69 is the results, please move this information to the results section.

Line 72. "Patients or patients". In my opinion this word does not need to be capital letter but please follow the same format.

Line 83: RMI: what is the maximum score of this scale? This index assesses mobility but what mobility? trunk mobility, lower limb mobility, general mobility? Please explain.

There is no explanation of the procedure or how the study was developed. There is a lack of information about what, when and how this study was conducted. Who were the supervisors? Were the patients alone during the assessments? or Were there any family members with them?

Table 1: the title of the table should be on top. Always add a footnote with the explanation of "y", "m", "s", (although we all know these meanings).

Line 230: (Table: 2): correct punctuation.

Line 239: same comment as in table 1.

Check writing and punctuation, take care of capital letters and full stops.

Author Response

Reviewer 2:

Comment 1:

Introduction: well explained and concise

Response to comment 1:

We thank the reviewer for the appreciation.

Comment 2:

Line 69 is the results, please move this information to the results section.

Response to comment 2:

We moved this information to the results section.

Comment 3:

Line 72. "Patients or patients". In my opinion this word does not need to be capital letter but please follow the same format.

Response to comment 3:

We adjusted the capitalization of letters to be more consistent.

Comment 4:

Line 83: RMI: what is the maximum score of this scale? This index assesses mobility but what mobility? trunk mobility, lower limb mobility, general mobility? Please explain.

Response to comment 4

We expanded the instrument description, including which type of mobility and the maximum score. (line 150 – 156)

Comment 5:

There is no explanation of the procedure or how the study was developed. There is a lack of information about what, when and how this study was conducted. Who were the supervisors? Were the patients alone during the assessments? or Were there any family members with them?

Response to comment 5:

1 - We added information about location, methodology, and supervisors of the study in the study design section.

 “This cross-sectional study was designed to find the various factors affecting mobility in patients with stroke and the influence of balance as a mediator on mobility. Assessment sessions were conducted before participants’ routine physical therapy appointments. Participants were assessed by trained physical therapists for the following: balance by Berg Balance Scale (BBS), functional mobility by Rivermead Mobility Index (RMI), gait speed by Timed Up and Go (TUG), and lower extremity strength by a handheld dynamometer.” (line 101-131).

2- Timeline of the study was added in the Participant’s section.

“Participants were enrolled in the study in the period of June 2019- December 2019.”               (line 140-141).

3- Patients were sometimes alone and other times with a family member. However, according to our inclusion criteria patients must be cognitively intact in order to sign the informed consent and to follow instructions of each assessment. Family members were not involved in any part of the study.

Comment 6:

Table 1: the title of the table should be on top. Always add a footnote with the explanation of "y", "m", "s", (although we all know these meanings).

Response to comment 6:

We adjusted tables titles and footnotes throughout the manuscript.

Comment 7:

Line 230: (Table: 2): correct punctuation.

Response to comment 7:

We corrected the punctuation error

Comment 8:

Line 239: same comment as in table 1.

Response to comment 8:

We adjusted the table’s title.

Comment 9:

Check writing and punctuation, take care of capital letters and full stops.

Response to comment 9:

We re-checked the manuscript for writing errors, and made necessary modifications.

Round 2

Reviewer 1 Report

The authors revised the articole according with the recommendations. 

Reviewer 2 Report

Dear authors,

Thank you to follow my suggestions.

Good job.